# Comparison of professionalism between emergency medicine resident physicians and faculty physicians: A multicenter cross-sectional study

Takashi Shiga[1,*], Yoshiyuki Nakashima[2], Yasuhiro Norisue[3], Tetsunori Ikegami[4], Takahiro Uechi[5], Yuhei Otaki[6], Hidehiko Nakano[7], Keibun Ryu[8], Shinjiro Wakai[9], Hiraku Funakoshi[3], Shigeki Fujitani[10], Yasuharu Tokuda[11]

1 Department of Emergency Medicine, International University of Health and Welfare Tokyo, Otawara, Japan, 2 Nakashima Clinic, Minato City, Tokyo, Japan, 3 Department of Emergency and Critical Care Medicine, Tokyo Bay Urayasu-Ichikawa Medical Center, Urayasu City, Chiba, Japan, 4 Department of Emergency Medicine, Kurashiki Central Hospital, Kurashiki City, Okayama, Japan, 5 Department of General Medicine, Kyorin University School of Medicine, Mitaka City, Tokyo, Japan, 6 Department of Emergency Medicine, Jikei University School of Medicine, Minato City, Tokyo, Japan, 7 Department of Emergency Medicine, Shonan Kamakura General Hospital, Kamakura City, Kanagawa, Japan, 8 Department of Emergency and Critical Care Medicine, Maebashi Red Cross Hospital, Maebashi City, Gunma, Japan, 9 Department of Emergency and Critical Care Medicine, Tokai University School of Medicine, Isehara City, Kanagawa, Japan, 10 Department of Emergency and Critical Care Medicine, St. Mariana University, Kawasaki City, Kanagawa, Japan, 11 Muribushi Okinawa for Teaching Hospitals, Urasoe City, Okinawa, Japan

☯ These authors contributed equally to this work.
* tshiga@iuhw.ac.jp

**Data Availability Statement:** The Ethics Committee of the Tokyo Bay Urayasu/Ichikawa Hospital (approval number:227) approved this

## Abstract

Professionalism is a critical competency for emergency medicine (EM) physicians, and professional behavior affects patient satisfaction. However, the findings of various studies indicate that there are differences in the interpretation of professionalism among EM resident physicians and faculty physicians. Using a cross-sectional survey, we aimed to analyze common challenges to medical professionalism for Japanese EM physicians and survey the extent of professionalism coursework completed during undergraduate medical education. We conducted a multicenter cross-sectional survey of EM resident physicians and faculty physicians at academic conferences and eight teaching hospitals in Japan using the questionnaire by Barry and colleagues. We analyzed the frequency of providing either the best or second-best answers to each scenario as the main outcome measure and compared the frequencies between EM resident physicians and EM faculty physicians. Fisher's exact test and the Wilcoxon rank sum test were used to analyze data. A total of 176 physicians (86 EM resident physicians and 90 EM faculty physicians) completed the survey. The response rate was 92.6%. The most challenging scenario presented to participants dealt with sexual harassment, and only 44.5% chose the best or second-best answers, followed by poor responses to the confidentiality scenario (69.9%). The frequency of either the best or second-best responses to the confidentiality scenario was significantly greater for EM resident physicians than for EM faculty physicians (77.1% versus 62.9%, p = 0.048). More

study, including provisions for data sharing. Section 9 ethical considerations mandate that there are restrictions on the availability of data due to the consent agreements for data security as well as the IRB approval, which allow access only to external researchers for research monitoring purposes. A non-author contact for requesting data access is as follows: official-website_bay@jadecom.jp.

**Funding:** The authors received no specific funding for this work.

**Competing interests:** The authors have declared that no competing interests exist.

participants in the EM resident physician group completed formal courses in medical professionalism than those in the EM faculty physician group (25.8% versus 5.5%, p < 0.01). Further, EM faculty physicians were less likely than EM resident physicians to provide acceptable responses in terms of confidentiality, and few of both had received professionalism training through school curricula. Continuous professionalism education focused on the prevention of sexual harassment and gender gap is needed for both EM resident physicians and faculty physicians in Japan.

## Introduction

Professionalism is a critical competency for physicians. Furthermore, professional behavior affects patient satisfaction.[1] The Accreditation Council for Graduate Medical Education (ACGME) and American Board of Internal Medicine (ABIM) regard professionalism as a way to accomplish a commitment to carry out professional responsibilities, adhere to ethical principles, and demonstrate sensitivity to a diverse patient population.[2] Teaching and measuring medical professionalism are sometimes challenging activities because of the several inherent contexts.[3–8] However, there are several scientific evidences that support the effectiveness of a systematic educational approach to medical professionalism.[9–12]

The specialty of emergency medicine (EM) is unique because shared decision-making and effective communication must take place in a short period. Thus, medical professionalism is critically important for EM physicians. However, 45% of EM program directors reported that two or more resident physicians have exhibited unprofessional behavior in their programs each year.[10] Several studies have pointed to differences in interpretations of professionalism among EM resident physicians,[10,11] and resident physicians describe role models as most influential for interpreting the meaning of professionalism.

The Barry Questionnaire is an assessment tool for evaluating views regarding professionalism; it is widely used in the US and Japan.[12–14] In a previous study, participants in the US performed better than participants in Japan in scenarios that were presented involving minor confidentiality and sexual harassment, but not for three scenarios (physician impairment, conflict of interest, and acceptance of gifts).[12] A recent study of novice physicians using the Barry Questionnaire mentions improvements in medical professionalism with respect to certain ethical challenges in Japan.[14]

To the best of our knowledge, there is no study citing a difference in medical professionalism between EM resident physicians and faculty physicians. Recognition of a gap in views regarding professionalism between EM resident physicians and faculty physicians will enable innovative curricular changes in EM postgraduate education.

Thus, by using a cross-sectional multicenter survey, we aimed to analyze responses regarding common challenges to medical professionalism for Japanese EM resident physicians and EM faculty physicians. Further, we surveyed the extent of education related to professionalism.

## Method

### Study design and setting

We conducted a multicenter cross-sectional study of EM resident physicians and EM faculty physicians in Japan using the Barry Questionnaire. Instead of mailing a survey to potential participants, we used existing hospital conferences or academic conferences held by Emergency Medicine Alliance, Japan, for administering the questionnaire. Those conferences were the regular staff conferences and were traditionally held on weekday mornings in most Japanese

teaching hospitals. Tokuda contacted ACGME for permission to use the Barry questionnaire (ACGME 2004). Permission was granted for translation and its use for the previous study.[12] The study was approved by the Institutional Review Board of Tokyo Bay Urayasu-Ichikawa Medical Center, Chiba, Japan (approval number: 227).

## Study population

The questionnaire was administered at eight geographically diverse tertiary care medical centers (three university hospitals and four community hospitals) and at biannual EM academic conferences held in 2017 by the Emergency Medicine Alliance, which is an organization designed to promote emergency medicine education by training emergency physicians and general internists. All participants were Japanese, and they were informed about the study, based on which they provided written consent prior to the survey. Participants were assured of confidentiality and anonymity.

## Study instrument

The Barry Questionnaire was developed and validated in a study conducted in Colorado (US) by Barry et al.[13] They performed the following steps to develop and evaluate the instrument. A scenario review was conducted by a panel of people with experience in medical ethics, clinical practice, or law; a consensus on the best response and second-best response for each scenario was derived. We have presented each scenario of the Barry Questionnaire with the best response and the second-best response in the supporting information. The Japanese version of the Barry Questionnaire was developed and implemented by Tokuda et al. in 2009.[12] In this previous study, content validity, cultural adaptation, and translation of the Japanese version of the instrument was confirmed by an independent panel comprised of physicians responsible for educational programs in participating hospitals. Their reference was the professionalism guideline of the Japanese Medical Association.

The questionnaire presents six challenging cases relevant to medical professionalism: acceptance of gifts, conflict of interest, confidentiality, physician impairment, sexual harassment, and honesty in documentation. Each scenario is followed by four or five possible responses.

After reviewing all six scenarios, participants were asked, "Have you ever experienced formal education in medical professionalism?" The question required a "yes" or "no" response. If the answer to this question was "yes," participants were then asked, "How many hours of coursework devoted to professionalism did you take?" and "Were you satisfied with the contents of these educational sessions? (yes or no)" For demographic information, we collected data regarding professional specialty, gender, and work status (resident or faculty physician) from each participant.

## Study outcomes

The primary outcome measure was the frequency with which EM resident physicians and EM faculty physicians provided either the best or second-best answers to each scenario. Secondary outcome measures were the frequencies of providing either the best or second-best answers to each scenario when stratifying participating physicians by gender or professionalism education completed as an undergraduate medical student.

## Statistical analysis

We analyzed the frequency of providing either the best or second-best answers to each scenario as the main outcome measure and compared frequencies between EM resident physicians and EM faculty physicians.

In addition, we analyzed the frequency of selecting either the best or second-best answer to each scenario and compared those frequencies when stratifying participating physicians by gender or professionalism education received as an undergraduate medical student.

Based on the results of a previous study by Tokuda et al., we projected that an observation of 72 physicians in each group would provide 80% capacity to detect a 20% decrease in the right responses (90% versus 70%).

Fisher's exact test and the Wilcoxon rank sum test were used to analyze data, where appropriate. Data were analyzed using Stata version 14 (College Station, TX). A two-tailed p-value of less than 0.05 was considered statistically significant.

## Results

During the study period, 176 EM physicians (86 EM resident physicians and 90 EM faculty physicians) completed the survey. The response rate was 92.6% (Fig 1). Overall, the median number of postgraduate years of the participants was six; further, 21.0% were females. The participants' characteristics over the study periods are shown in Table 1. The ratio of learning experiences related to medical professionalism through school curricula was significantly higher for EM resident physicians than EM faculty physicians (25.8% vs. 5.5%; p < 0.01). Of 176 participants, 25 (14%) participants (20 EM resident physicians and 5 faculty physicians) reported completing formal educational courses about medical professionalism. Among 25 participants, the median hours for coursework devoted to professionalism were three hours (the range was 1–30 hours) of the entire school curricula. Of these 25 participants, six (24%) participants (including five resident physicians) reported that they were satisfied with these educational sessions.

Table 2 shows participants' characteristics according to the institution of participating physicians. For each institution, we report a number of eligible faculty/resident physicians as well as participating faculty/resident physicians.

Table 3 compares the frequencies with which study participants (EM resident physicians and EM faculty physicians) provided the best or second-best responses. Linear plot graphs correlating response to each scenario versus PGY were provided in supporting information (S2 -S8). The frequency of either the best or second-best responses to the confidentiality scenario was significantly greater for EM resident physicians than for EM faculty physicians (77.1% vs. 62.9%; p = 0.048). The most challenging case for all participating physicians was selecting the best or second-best responses to the sexual harassment scenario (i.e., frequency was 44.5%). For the confidentiality scenario, the frequency of best or second-best responses was 69.9%. In the sexual harassment scenario, no physician chose the worst response (i.e., "Do nothing, on the basis that the faculty member was simply showing his appreciation for a job well done").

Table 4 shows the frequencies of the best or second-best responses when stratified by participants' gender. There were no differences in the responses for each scenario between male and female physicians.

Table 5 compares the frequencies of best or second-best responses for EM physicians who had been educated in medical professionalism through their school curricula and those without any education regarding professionalism in their school curricula. There were no significant differences in responses for each scenario.

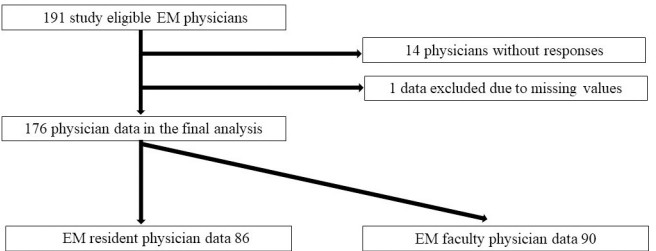

**Fig 1. Patient flow in the study (EM: Emergency medicine).**

## Discussion

To the best of our knowledge, this is the first study that directly compares views regarding medical professionalism of EM resident physicians and faculty physicians. In this multicenter study of 176 EM physicians, we found that views regarding medical professionalism of EM resident physicians were superior to those of EM faculty physicians for the confidentiality scenario. A substantial proportion of Japanese EM resident physicians and faculty physicians has not received professionalism education in medical school. However, in our analysis, the effectiveness of medical professionalism training in undergraduate medical education was not confirmed. Our findings are useful for recognizing the necessity of an effective education in medical professionalism in the field of EM.

Tokuda et al. showed that the Japanese physicians were unable to respond acceptably to challenges to professionalism, particularly concerning sexual harassment, honesty, and confidentiality.[12] Similar to the present study, their study showed that the professionalism of resident physicians was superior in terms of the confidentiality scenario. In addition, Kinoshita et al. reported improved responses to the Barry Questionnaire by Japanese physicians.[14] The study by Barry et al.[13] indicates that satisfaction with training in professionalism is significantly related to the amount of relevant coursework. Findings of the present study are consistent with those of the prior studies and extends them by demonstrating the further need for improvement in medical professionalism education for Japanese physicians in terms of confidentiality and sexual harassment.

A substantial proportion of EM physicians failed to provide acceptable responses to the challenges to professionalism in several scenarios. Particularly, more than half of the participants did not respond suitably to the sexual harassment scenario. Two-thirds of the participants selected an unfavorable choice or improper response to the sexual harassment scenario (i.e., ask the resident if the gesture made her uncomfortable). This response might reflect Japan's unique culture in which the Japanese tend to hide their emotions from others. Many

**Table 1. Participants' characteristics according to physician work status.**

| Variables | Total | Resident | Faculty | p-value |
|---|---|---|---|---|
| | N = 176 | n = 86 | n = 90 | |
| PGY, median (IQR) | 6 (5–11) | 5 (3–5) | 10 (8–16) | <0.01 |
| Male, n (%) | 139 (79.0) | 60 (69.8) | 79 (86.8) | <0.01 |
| Community hospital, n (%) | 119 (67.6) | 62 (72.0) | 57 (63.3) | 0.26 |
| Professionalism education in medical school, n (%) | 25 (14.2) | 19 (25.8) | 5 (5.5) | <0.01 |

Abbreviations: PGY, postgraduate year; IQR, interquartile range

Wilcoxon rank-sum test for continuous variables, Fisher's exact test for categorical variables

**Table 2. Participants' characteristics according to institution.**

| Institution | Eligible physicians | | | Participating physicians | | |
|---|---|---|---|---|---|---|
| | Resident | Faculty | Institutional | Resident | Faculty | Institutional |
| | n = 98 | n = 93 | n = 191 | n = 86 | n = 90 | N = 176 |
| A | 8 | 9 | 17 | 8 | 9 | 17 |
| B | 9 | 9 | 18 | 3 | 9 | 12 |
| C | 6 | 6 | 12 | 6 | 6 | 12 |
| D | 16 | 8 | 24 | 16 | 8 | 24 |
| E | 6 | 10 | 16 | 4 | 9 | 13 |
| F | 6 | 6 | 12 | 5 | 6 | 11 |
| G | 9 | 7 | 16 | 8 | 7 | 15 |
| H | 2 | 4 | 6 | 2 | 4 | 6 |
| Academic conferences | 36 | 34 | 70 | 34 | 32 | 66 |

Japanese think that this trait is admirable. Based on this cultural context, many Japanese participants might have thought that the female in the hypothetical case did not express her discomfort through her facial expressions intentionally. Thus, they could have chosen to request confirmation from her to be accurate. Further, frequencies of the best or second-best answers for the scenarios concerning confidentiality were also relatively low (69.9%). As seen in the study by Tokuda in 2009, EM faculty physicians' responses were inferior to those of the resident physicians for the confidentiality scenario (77.1% vs. 62.9%, p = 0.048). There is a possibility that faculty physicians may portray decreased sensitivity either as a result of increased experiences or burnout. Compared to Tokuda et al.'s study, feedback from EM faculty physicians in the current study regarding responses to gifts scenario showed little difference (87.9% versus 89.8%, P = 0.81). This data might reflect gradual dissemination of professionalism education in Japan. Overall, our findings may reflect a lack of evidenced based professionalism education for EM faculty physicians during their training phase. A growing body of evidence indicates the utilization of guided reflection and formative feedback in professional identity formation.[15, 16] Further, there has been an increased focus on a collaborative learning environment.[16] In addition, there is a strong concern regarding gender gap in academic activities in the Japanese medical field.[17] As a society, we are must work towards reducing gender gap.

This study has several educational implications. First, low performance in the sexual harassment scenario represents a significant problem in the Japanese medical field. One Japanese study reported that 58.6% of female resident physicians had experienced sexual harassment.

**Table 3. Frequency of the best or second-best responses for scenarios presented by residents and faculty.**

| Scenario | Total | Resident | Faculty | p-value |
|---|---|---|---|---|
| | n = 176 | n = 86 | n = 90 | |
| Gifts, n (%) | 153 (88.9) | 73 (87.9) | 80 (89.8) | 0.81 |
| Conflict of interest, n (%) | 154 (89.7) | 72 (86.7) | 82 (92.2) | 0.32 |
| Confidentiality, n (%) | 121 (69.9) | 64 (77.1) | 56 (62.9) | 0.048 |
| Impairment, n (%) | 145 (84.3) | 67 (80.7) | 78 (87.6) | 0.29 |
| Harassment, n (%) | 77 (44.5) | 36 (43.3) | 41 (46.1) | 0.76 |
| Honesty, n (%) | 151 (87.3) | 76 (81.7) | 74 (83.1) | 0.11 |

Abbreviation: EM, Emergency medicine

Fisher's exact test for categorical variables

**Table 4. Frequency of the best or second-best responses according to gender for the scenarios presented.**

| Scenario | Male | Female | p-value |
|---|---|---|---|
| | n = 139 | n = 37 | |
| Gifts, n (%) | 123 (89.1) | 30 (88.2) | 1 |
| Conflict of interest, n (%) | 123 (89.1) | 32 (91.4) | 1 |
| Confidentiality, n (%) | 92 (66.7) | 28 (82.3) | 0.1 |
| Impairment, n (%) | 117 (84.8) | 28 (82.3) | 0.79 |
| Harassment, n (%) | 61 (44.2) | 16 (47.1) | 0.85 |
| Honesty, n (%) | 120 (87.0) | 30 (88.2) | 1 |

Fisher's exact test for categorical variables

[18] Harassment and discrimination in medical training is recognized internationally.[19] Furthermore, the emergency department is one of the highest risk areas for abuse and harassment in the hospital setting.[20] In Japan, measures against sexual harassment at work are mandated by the Japanese law.[21] Based on our results, we can confirm that the sexual harassment policy of each emergency department needs to be reviewed and implemented with continuous professional education, strict reporting procedures, and counseling for victims and witnesses. We also need to provide more pre- and postgraduate education on sexual harassment. Second, low performance with regard to confidentiality, especially among EM faculty physicians, is another problem. Compared with results from the previous study by Tokuda et al.[12] 10 years prior, responses by EM faculty physicians to the confidentiality scenario in the present study were lower by 10%. This finding suggests the need for continuous education on professionalism after residency training in addition to rigorous pre-graduate education on professionalism.

## Limitations

Our study has several potential limitations. First, we need to confirm the applicability of the Barry Questionnaire to the Japanese context. To this aspect, Tokuda et al. published the validation study of the Barry Questionnaire after engaging in robust processes. However, this previous study did not entirely follow standards of translation and cultural adaptation as recommended in translation guidelines in detail.[12] Second, there is no study that examines whether it is suitable to use the Barry Questionnaire to evaluate EM physicians' professionalism. However, there has not been a well-validated tool to specifically investigate EM

**Table 5. Frequency of best or second-best responses for the scenarios presented according to undergraduate professionalism education.**

| Scenario | Undergraduate Professionalism Education | | |
|---|---|---|---|
| | Yes | No | p-value |
| | n = 25 | n = 148 | |
| Gifts, n (%) | 21 (84.0) | 132 (89.2) | 0.73 |
| Conflict of interest, n (%) | 21 (84.0) | 135 (90.6) | 0.3 |
| Confidentiality, n (%) | 21 (84.0) | 100 (67.5) | 0.16 |
| Impairment, n (%) | 21 (84.0) | 125 (84.5) | 1 |
| Harassment, n (%) | 9 (36.0) | 68 (46.0) | 0.39 |
| Honesty, n (%) | 24 (96.0) | 127 (85.8) | 0.21 |

Fisher's exact test for categorical variables

physicians' professionalism to date.[9] Therefore, we used the Barry Questionnaire for our study. Third, the case scenario approach is not the only method for measuring attitudes about professionalism. It only addresses the cognitive aspects of professionalism. It could be better to combine another method that addresses professional behaviors as well as cognitive aspects.[22] Assessment of professionalism could be performed through subjective, narrative, and personal approaches.[23] Fourth, there is the possibility of bias in the selection of participants in our research. However, the seven hospitals were in geographically diverse areas of Japan. All hospitals were teaching hospitals and tertiary care medical centers. In addition, we intentionally set the balance between community hospitals and academic hospitals to ensure diversity among participants.

## Conclusion

Compared with EM resident physicians, EM faculty physicians were less likely to respond acceptably regarding matters of confidentiality. Few EM faculty physicians had been educated about professionalism during their undergraduate years. Furthermore, both EM resident physicians and faculty physicians did not provide acceptable responses regarding harassment. Continuous education about professionalism focused on the prevention of sexual harassment and gender gap is needed in medical education for both EM resident physicians and EM faculty physicians in Japan.

## Supporting information

**S1 File. Barry Questionnaire for professionalism.**
(DOCX)

**S1 Fig. Linear plot graph to show correlation between response to a scenario regarding gifts and post graduate year.**
(TIF)

**S2 Fig. Linear plot graph to show correlation between response to a scenario regarding conflict of interest and post graduate year.**
(TIF)

**S3 Fig. Linear plot graph to show correlation between response to a scenario regarding confidentiality and post graduate year.**
(TIF)

**S4 Fig. Linear plot graph to show correlation between response to a scenario regarding impairment and post graduate year.**
(TIF)

**S5 Fig. Linear plot graph to show correlation between response to a scenario regarding harassment and post graduate year.**
(TIF)

**S6 Fig. Linear plot graph to show correlation between response to a scenario regarding honesty and post graduate year.**
(TIF)

**S7 Fig. Linear plot graph to show correlation between responses to all scenarios and post graduate year.**
(TIF)

## Acknowledgments

We would like to thank all of the physicians who participated in this study.

## Author Contributions

**Conceptualization:** Takashi Shiga, Yoshiyuki Nakashima, Yuhei Otaki.

**Data curation:** Takashi Shiga, Tetsunori Ikegami, Takahiro Uechi, Hidehiko Nakano, Keibun Ryu, Shinjiro Wakai, Hiraku Funakoshi.

**Formal analysis:** Takashi Shiga.

**Methodology:** Takashi Shiga, Yoshiyuki Nakashima, Yasuhiro Norisue, Hiraku Funakoshi.

**Project administration:** Takashi Shiga, Yoshiyuki Nakashima.

**Software:** Takashi Shiga.

**Supervision:** Yasuhiro Norisue, Shigeki Fujitani, Yasuharu Tokuda.

**Writing – original draft:** Takashi Shiga.

**Writing – review & editing:** Takashi Shiga, Yoshiyuki Nakashima, Hiraku Funakoshi, Shigeki Fujitani.

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
