## [Decision Letter · Decision Letter 0]

2 Dec 2019

PONE-D-19-17887

Comparison of professionalism between emergency medicine resident physicians and faculty physicians: A multicenter cross-sectional study

PLOS ONE

Dear Dr. Shiga,

Thank you for submitting your manuscript to PLOS ONE. After careful consideration, we feel that it has merit but does not fully meet PLOS ONE’s publication criteria as it currently stands. Therefore, we invite you to submit a revised version of the manuscript that addresses the points raised during the review process.

We would appreciate receiving your revised manuscript by Jan 16 2020 11:59PM. To enhance the reproducibility of your results, we recommend that if applicable you deposit your laboratory protocols in protocols.io, where a protocol can be assigned its own identifier (DOI) such that it can be cited independently in the future. For instructions see: http://journals.plos.org/plosone/s/submission-guidelines#loc-laboratory-protocols

We look forward to receiving your revised manuscript.

Kind regards,

Cesario Bianchi

Academic Editor

PLOS ONE

Additional Editor Comments:

Dear Dr Shiga,

Your manuscript was reviewed by 2 experts that found the work interesting. They have, however, some important comments and suggestions to help you improve your manuscript. Please answer all and every comment. Make changes in the the revised version according to the reviewers comments, if you find necessary, and return it as soon as possible.

thank you

2. Please provide additional details regarding participant consent.

In the ethics statement in the Methods and online submission information, please ensure that you have specified (1) whether consent was informed and (2) what type you obtained (for instance, written or verbal, and if verbal, how it was documented and witnessed).

Reviewers' comments:

Reviewer's Responses to Questions

**Comments to the Author**

1. Is the manuscript technically sound, and do the data support the conclusions?

Reviewer #1: Yes

Reviewer #2: Partly

2. Has the statistical analysis been performed appropriately and rigorously? 

Reviewer #1: Yes

Reviewer #2: Yes

3. Have the authors made all data underlying the findings in their manuscript fully available?

Reviewer #1: Yes

Reviewer #2: No

4. Is the manuscript presented in an intelligible fashion and written in standard English?

Reviewer #1: Yes

Reviewer #2: Yes

5. Review Comments to the Author

Reviewer #1: Dr. Shiga and colleagues present a novel study evaluating professionalism between emergency medicine residents and faculty physicians in Japan. The topic of the manuscript is timely as professionalism and communication are increasingly recognized as essential to patient care and satisfaction. As the authors have described, the provision of formal education regarding professionalism is not something which was routinely part of medical school curricula in the past as evidenced by the lower percentage of faculty who had received formal training. Interestingly, the residents provided the best or second best answer more frequently than the attending group. The primary strengths of the manuscript include:

1) A timely topic

2) Use of a cross-sectional survey across multiple centers

3) High response rate to the survey

4) Use of the validated Barry survey

The following would enhance the manuscript:

1) The authors note 45% of EM program directors reported 2 or more resident physicians had exhibited unprofessional behavior over the prior year (Ref 10). It would be helpful if the authors provided data regarding whether the residents and faculty in the study had any reports of unprofessional behavior and if there was a correlation between these reports and scores on the Barry survey.

2) Additional discussion regarding whether Barry scores truly relate to quality of care / patient satisfaction would be helpful. Are there clinical evaluations of the participants available to compare with their scores on the Barry survey?

3) Additional discussion regarding why faculty performed worse than residents would be helpful. Is it only a lack of formal training? Is there a decrease in sensitivity with increased experience / burnout?

4) A linear plot correlating scores in each area vs PGY year would be interesting to see how time / experience effect Barry scores in each area

Reviewer #2: 1. Is the manuscript technically sound, and do the data support the conclusions?

The manuscript deals with an important and complex issue in Medical Education: professionalism development among Emergency Medicine physicians. However, there major methodological flaws that authors should take into account when submitting a revised version of the manuscript:

1. The Barry Questionnaire does not measure professional competence, as stated in many parts of the manuscript (ie, Introduction – lines 104, 105; Discussion – line 221). Authors should bear this limitation in mind when stating the study purposes and conclusions.

2. It is not clear whether participants were recruited at the hospitals or at scientific/academic meetings held at specific places (what do authors mean by “biannual EM academic meetings – is that a conference/congress?). How many EM residents and faculty physicians are there in the seven hospitals included in the study? These are important issues that may impair conclusions and generalization of results.

3. Data on the translation, cultural adaptation and validation of the Barry Questionnaire to the Japanese cultural is not provided (the study referenced in the Methods section does not provide information on translation and adaptation techniques, reliability and validity of the questionnaire).

4. Results on professionalism education and harassment should be discussed in the light of existing evidence not only from Japan, but also from international studies. What kind of educational interventions work best in developing professionalism among residents and physicians?

5. Previous results on Japanese residents and physicians’ responses to challenging scenarios related to medical professionalism should be revised. The paper from Tokuda et al. (2009) (Reference 12) shows conflicting results concerning the dimensions “Gifts” and “Confidentiality”: faculty physicians performed better than residents in that study sample. What are the possible explanations to such differences?

6. Authors should also bear in mind the possibility of gender bias in their results, particularly concerning participants’ responses on “harassment”. Are there any cultural issues related to gender roles in Japan that may explain the poor performance of residents and faculty physicians in this scenario?

2. Has the statistical analysis been performed appropriately and rigorously?

Yes, although sample size calculation and participation rate (considering the total number of EM residents and physicians among all seven hospitals included in the study) should be clearly stated in the manuscript.

3. Does the manuscript adhere to the PLOS Data Policy?

Data do not seem to be public.

4. Is the manuscript presented in an intelligible fashion and written in standard English?

Yes, although professional English editing is necessary.

Minor revisions:

1. Abstract

- Line 58: include number of participating hospitals

- Lines 74 – 75: results do not support this statement, since participants had properly responded to most of the scenarios presented, except for “harassment”.

2. Method:

- Lines 118 – 119: it is not clear how permission to use the questionnaire was granted

- Line 127: was consent written?

3. Discussion:

- Lines 251 – 253: authors should avoid repetition of results and should present a deeper discussion of the study main results, as I mentioned in comments 4 and 6 above (major revisions)

6. PLOS authors have the option to publish the peer review history of their article (what does this mean?). If published, this will include your full peer review and any attached files.

Reviewer #1: No

Reviewer #2: Yes: Helena B M S Paro

---

## [Author Response · Author response to Decision Letter 0]

5 Jan 2020

Dear Dr. Cesario Bianchi, 

Thank you very much for your ongoing consideration of our manuscript ‘PONE-D-19-17887 Comparison of professionalism between emergency medicine resident physicians and faculty physicians: A multicenter cross-sectional study’ for publication in the PLOS ONE. Below, we address point-by-point responses to comments.

We remain very enthusiastic about publishing our original scientific article in the Journal and look forward to your editorial decision.

Sincerely,

Takashi Shiga, M.D., M.P.H.　

(on behalf of all authors)

Response:

We appreciate the editor’s kind suggestion. Upon submission regarding this revision, we have reviewed the requirements and believe that our manuscript is consistent with the journal requirements.

2. Please provide additional details regarding participant consent.

In the ethics statement in the Methods and online submission information, please ensure that you have specified (1) whether consent was informed and (2) what type you obtained (for instance, written or verbal, and if verbal, how it was documented and witnessed).

Response:

We appreciate editor’s insightful comment. Consent was informed and obtained as written consent. The manuscript was updated accordingly (Page 8, line 134). 

3. We note that you have indicated that data from this study are available upon request. 

Response:

The Ethics Committee of the Tokyo Bay Urayasu/Ichikawa Medical Center (approval number: 227) approved this study, including provisions for data sharing. Section 9 ethical considerations mandate that there are restrictions on the availability of data due to the consent agreements for data security as well as the IRB approval, which allow access only to external researchers for research monitoring purposes. A non-author contact for requesting data access is as follows: official-website_bay@jadecom.jp.

Reviewers' comments:

Reviewer #1: 

The following would enhance the manuscript:

1) The authors note 45% of EM program directors reported 2 or more resident physicians had exhibited unprofessional behavior over the prior year (Ref 10). It would be helpful if the authors provided data regarding whether the residents and faculty in the study had any reports of unprofessional behavior and if there was a correlation between these reports and scores on the Barry survey.

Response:

We appreciate reviewer’s insightful comments. It is ideal to have reports of unprofessional behaviors of study participants. However, unfortunately current dataset does not include such information. Because it was an anonymous survey, it is not possible to conduct additional inquiry to the participants.

2) Additional discussion regarding whether Barry scores truly relate to quality of care/ patient satisfaction would be helpful. Are there clinical evaluations of the participants available to compare with their scores on the Barry survey?

Response:

We appreciate reviewer’s insightful comments. It is ideal to have clinical evaluations of study participants. However, unfortunately current dataset does not include such information. Because it was an anonymous survey, it is not possible to conduct additional inquiry to the participants.

3) Additional discussion regarding why faculty performed worse than residents would be helpful. Is it only a lack of formal training? Is there a decrease in sensitivity with increased experience / burnout?

Response:

We appreciate reviewer’s insightful comments. It is possible for the faculty physicians to have a decrease in sensitivity with increased experience/burnout. We have added this possibility to our manuscript (Page 19, line 273).

4) A linear plot correlating scores in each area vs PGY year would be interesting to see how time / experience effect Barry scores in each area.

Response:

We appreciate reviewer’s insightful comments. We have created supporting information files to show the linear plot graphs (S2-S7). In addition, we created a linear plot to show correlation between total number of right answers and PGY (S8). (Page 13, line 210)

Reviewer #2:

1. The Barry Questionnaire does not measure professional competence, as stated in many parts of the manuscript (ie, Introduction – lines 104, 105; Discussion – line 221). Authors should bear this limitation in mind when stating the study purposes and conclusions.

Response:

We appreciate reviewer’s insightful comments. In the revised manuscript, we tried to avoid the use of expression ‘professional competence’(Page 6 line 106 and Page 17 Line 238). 

2-1. It is not clear whether participants were recruited at the hospitals or at scientific/academic meetings held at specific places (what do authors mean by “biannual EM academic meetings – is that a conference/congress?). 

Response:

We appreciate reviewer’s insightful comments. Out of 176 participants, 66 physicians were recruited at the Emergency Medicine academic conferences for emergency medicine physician/general internist. We have updated the manuscript related to this section accordingly. (Page 8 line 131, page 13 line 206, and table 2) 

2-2. How many EM residents and faculty physicians are there in the seven hospitals included in the study? These are important issues that may impair conclusions and generalization of results.

Response:

We appreciate reviewer’s insightful comments. We have provided the data regarding demographic information of the participating physicians in table 2. (Page 13, line 206)

3. Data on the translation, cultural adaptation and validation of the Barry Questionnaire to the Japanese cultural is not provided (the study referenced in the Methods section does not provide information on translation and adaptation techniques, reliability and validity of the questionnaire).

Response:

We appreciate reviewer’s insightful comments. The experts of professionalism, Tokuda Y, Barnett PB, Norisue Y, Konishi R, Kudo H, Miyagi S have discussed and assured the content validity as well as cultural adaptation to the Japanese context upon conduction of the study in 2009. (Questionnaire survey for challenging cases of medical professionalism in Japan. Med Teach. 2009;31(6):502-507.) We have updated the manuscript accordingly (Page 9, line 146). 

4. Results on professionalism education and harassment should be discussed in the light of existing evidence not only from Japan, but also from international studies. What kind of educational interventions work best in developing professionalism among residents and physicians?

Response:

We appreciate reviewer’s insightful comments. We have cited articles by Wald et al as well as Goldie to reinforce discussion regarding educational interventions such as reflection, formative feedback and collaborative learning environment in developing professionalism in medical education (Page 19, line 280).

5. Previous results on Japanese residents and physicians’ responses to challenging scenarios related to medical professionalism should be revised. The paper from Tokuda et al. (2009) (Reference 12) shows conflicting results concerning the dimensions “Gifts” and “Confidentiality”: faculty physicians performed better than residents in that study sample. What are the possible explanations to such differences?

Response:

We appreciate reviewer’s insightful comments. In the previous study by Tokuda et al. (2009) showed better response by faculty physicians in terms of gift (55.0% versus 90.4%, P<0.01). In the current study, the difference was quite small (87.9% versus 89.8%, P=0.81). This data might reflect gradual dissemination of professionalism education in Japan (Page 19, line 274). 

In terms of confidentiality, the previous study showed better response by resident physicians in terms of confidentiality (90.0% versus 68.7%, P=0.001). In the current study, we have observed similar but smaller difference (77.1% versus 62.9%, P=0.048). (Page 19, line 271)

6. Authors should also bear in mind the possibility of gender bias in their results, particularly concerning participants’ responses on “harassment”. Are there any cultural issues related to gender roles in Japan that may explain the poor performance of residents and faculty physicians in this scenario?

Response:

We appreciate reviewer’s insightful comments. As pointed, there is strong concern regarding gender role/gap in the Japanese medical field. We have included an additional sentence quoting relevant article regarding this concern (Page 19, line 282).

7. Sample size

Sample size calculation and participation rate (considering the total number of EM residents and physicians among all seven hospitals included in the study) should be clearly stated in the manuscript.

Response: 

We appreciate reviewer’s insightful comments. Based on the results of previous study by Tokuda et al, we calculated that the observation of 72 physicians in each group would provide 80% power to detect a 20% decrease in the right responses (90% vs 70%) (Page 11, line 177).

Minor revisions:

1. Abstract

- Line 58: include number of participating hospitals

- Lines 74 – 75: results do not support this statement, since participants had properly responded to most of the scenarios presented, except for “harassment”.

Response:

We appreciate reviewer’s insightful comments. 

-Line 58: we have included the number of participating hospitals in the abstract (Page 3, line 58).

-Lines 74-75: we have changed the expression as ‘Continuous professionalism education emphasizing prevention of sexual harassment and gender gap is needed’ (Page 4, line 74).

2. Method:

- Lines 118 – 119: it is not clear how permission to use the questionnaire was granted

- Line 127: was consent written?

Response:

We appreciate reviewer’s insightful comments. 

-Tokuda contacted with ACGME regarding permission of the Barry’s questionnaire and it was granted for translation and use for the previous study (Page 7, line 121). 

-Written consent was obtained from participants (Page 8, line 134).

3. Discussion:

- Lines 251 – 253: authors should avoid repetition of results and should present a deeper discussion of the study main results, as I mentioned in comments 4 and 6 above (major revisions)

Response:

We appreciate reviewer’s insightful comments. As described above in the sections of comments 4 and 6, we have revised previous lines 251-253 accordingly (Page 19, line 280).

---

## [Decision Letter · Decision Letter 1]

28 Jan 2020

PONE-D-19-17887R1

Comparison of professionalism between emergency medicine resident physicians and faculty physicians: A multicenter cross-sectional study

PLOS ONE

Dear Dr. Shiga,

Thank you for submitting your manuscript to PLOS ONE. After careful consideration, we feel that it has merit but does not fully meet PLOS ONE’s publication criteria as it currently stands. Therefore, we invite you to submit a revised version of the manuscript that addresses the points raised during the review process.

Please address the concerns raised by reviewer#2 and make changes, if you find appropriated, to the revised manuscript. Thank you

We would appreciate receiving your revised manuscript by Mar 13 2020 11:59PM. To enhance the reproducibility of your results, we recommend that if applicable you deposit your laboratory protocols in protocols.io, where a protocol can be assigned its own identifier (DOI) such that it can be cited independently in the future. For instructions see: http://journals.plos.org/plosone/s/submission-guidelines#loc-laboratory-protocols

We look forward to receiving your revised manuscript.

Kind regards,

Cesario Bianchi

Academic Editor

PLOS ONE

Additional Editor Comments (if provided):

Dear Dr. Shiga,

Thank you for carefully revising your submission. I need, however, that you address some concerns raised by Reviewer#2 before I make a final decision.

Thank you for your

Reviewers' comments:

Reviewer's Responses to Questions

**Comments to the Author**

1. If the authors have adequately addressed your comments raised in a previous round of review and you feel that this manuscript is now acceptable for publication, you may indicate that here to bypass the “Comments to the Author” section, enter your conflict of interest statement in the “Confidential to Editor” section, and submit your "Accept" recommendation.

Reviewer #2: All comments have been addressed

2. Is the manuscript technically sound, and do the data support the conclusions?

Reviewer #2: (No Response)

3. Has the statistical analysis been performed appropriately and rigorously? 

Reviewer #2: Yes

4. Have the authors made all data underlying the findings in their manuscript fully available?

Reviewer #2: No

5. Is the manuscript presented in an intelligible fashion and written in standard English?

Reviewer #2: No

6. Review Comments to the Author

Reviewer #2: There are still some mistaken expressions to the attitudes towards towards professionalism as measured by the questionnaire ("medical professionalism is used in line 237, eg). General English editing is necessary: there is a typing mistake in line 265 ("facial expressions); "focus ON collaborative learning" (line 278).

The translation and validation process of the questionnaire cited in Ref 12 did not follow rigorous standards of translation and cultural adaptation as recommended in various translation guidelines. Authors should point it as a limitation to the study design.

7. PLOS authors have the option to publish the peer review history of their article (what does this mean?). If published, this will include your full peer review and any attached files.

Reviewer #2: Yes: Helena BMS Paro

---

## [Author Response · Author response to Decision Letter 1]

7 Feb 2020

Reviewer #2: 

1) There are still some mistaken expressions to the attitudes towards professionalism as measured by the questionnaire ("medical professionalism is used in line 237, eg). 

Response:

We appreciate reviewer’s insightful comments. In the revised manuscript, we tried to use ‘views regarding medical professionalism’ (Page 17 line 240). 

2) General English editing is necessary: there is a typing mistake in line 265 ("facial expressions); "focus ON collaborative learning" (line 278).

Response:

We appreciate reviewer’s insightful comments. We have updated as ‘facial expressions’ (Line 268) and focus on collaborative learning (Line 281) in each section accordingly. 

3) The translation and validation process of the questionnaire cited in Ref 12 did not follow rigorous standards of translation and cultural adaptation as recommended in various translation guidelines. Authors should point it as a limitation to the study design.

Response:

We appreciate reviewer’s insightful comments. We have updated this point at the limitation section (Page21 line 307).

Journal requirement:

While revising your submission, please upload your figure files to the Preflight Analysis and Conversion Engine (PACE) digital diagnostic tool, https://pacev2.apexcovantage.com/. PACE helps ensure that figures meet PLOS requirements.

Response:

We appreciate journal officer’s instructions. We have checked our images through PACE system and uploaded to the submission system accordingly.

---

## [Editor Report · Decision Letter 2]

25 Feb 2020

Comparison of professionalism between emergency medicine resident physicians and faculty physicians: A multicenter cross-sectional study

PONE-D-19-17887R2

Dear Dr. Shiga,

We are pleased to inform you that your manuscript has been judged scientifically suitable for publication and will be formally accepted for publication once it complies with all outstanding technical requirements.

With kind regards,

Cesario Bianchi

Academic Editor

PLOS ONE

Additional Editor Comments (optional):

Dear Dr Shiga,

Thank you for additional and careful revision. I find your manuscript (revision 2) acceptable for publication at this time.
---

## [Editor Report · Acceptance letter]

28 Feb 2020

PONE-D-19-17887R2 

Comparison of professionalism between emergency medicine resident physicians and faculty physicians: A multicenter cross-sectional study 

Dear Dr. Shiga:

I am pleased to inform you that your manuscript has been deemed suitable for publication in PLOS ONE. Congratulations! Your manuscript is now with our production department. 

With kind regards,

on behalf of

Dr. Cesario Bianchi 

Academic Editor

PLOS ONE